# DeSCo: Towards Scalable Deep Subgraph Counting

## Abstract

Subgraph counting is the problem of determining the number of a given query graph in a large target graph. Despite being a #P problem, subgraph counting is a crucial graph analysis method in domains ranging from biology and social science to risk management and software analysis. However, existing exact counting methods take combinatorially long runtime as target and query sizes increase. Existing approximate heuristic methods and neural approaches fall short in accuracy due to high label dynamic range, limited model expressive power, and inability to predict the distribution of subgraph counts in the target graph. Here we propose DeSCo, a neural deep subgraph counting framework, which aims to accurately predict the count and distribution of query graphs on any given target graph. DeSCo uses *canonical partition* to divide the large target graph into small neighborhood graphs and predict the canonical count objective on each neighborhood. The proposed partition method avoids missing or double-counting any patterns of the target graph. A novel *subgraph-based heterogeneous graph neural network* is then used to improve the expressive power. Finally, *gossip correction* improves counting accuracy via prediction propagation with learnable weights. Compared with state-of-the-art approximate heuristic and neural methods. DeSCo achieves $437\times$ improvement in the mean squared error of count prediction and benefits from the polynomial runtime complexity.

## 1 Introduction

Given a *query* graph and a *target* graph, the problem of subgraph counting is to count the number of *patterns*, defined as subgraphs of the target graph, that are graph-isomorphic to the query graph Ribeiro et al. (2021).

Subgraph counting is crucial for domains including biology Takigawa & Mamitsuka (2013); Solé & Valverde (2008); Adamcsek et al. (2006); Bascompte & Melián (2005); Bader & Hogue (2003), social science Uddin et al. (2013); Prell & Skvoretz (2008); Kalish & Robins (2006); Wasserman et al. (1994), risk management Ribeiro et al. (2017); Akoglu & Faloutsos (2013), and software analysis Valverde & Solé (2005); Wu et al. (2018).

While being an essential method in graph and network analysis, subgraph counting is a #P-complete problem Valiant (1979). Due to the computational complexity, existing exact counting algorithms are restricted to small query graphs with no more than 5 vertices Pinar et al. (2017); Ortmann & Brandes (2017); Ahmed et al. (2015). The commonly used VF2 Cordella et al. (2004) algorithm fails to even count a single query of 5-node chain within a week's time budget on a large target graph Astro Leskovec et al. (2007) with nineteen thousand nodes.

Luckily, approximate counting of query graphs is sufficient in many real-world use cases Iyer et al. (2018); Kashtan et al. (2004); Ribeiro & Silva (2010). Approximation methods can scale to large targets by substructure sampling, random walk, and color-based sampling, allowing estimation of the frequency of query graph occurrences. Very recently, Graph Neural Networks (GNNs) are employed as a deep learning-based approach to subgraph counting Zhao et al. (2021); Liu et al. (2020); Chen et al. (2020). The target graph and the query graph are embedded via a GNN, which predicts the motif count through a regression task.

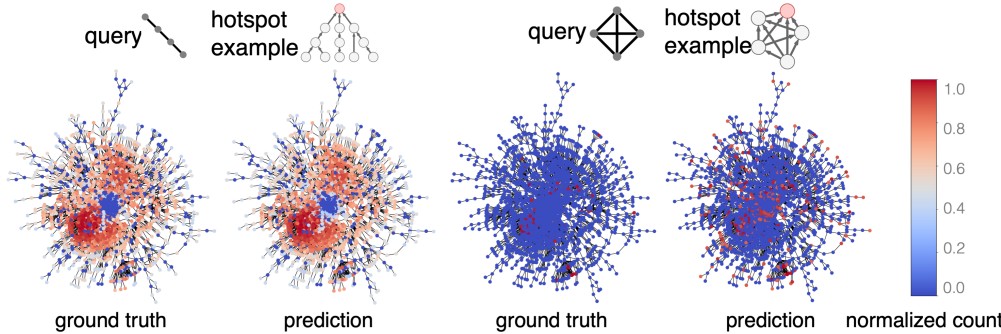

Figure 1: The ground truth and predicted count distributions of different query graphs over the target graph CiteSeer, a citation network. The hotspots are where the patterns appear most often in the target graph. The hotspots of k-chains represent overlapped linear citation chains, indicating original publications that motivate multiple future directions of incremental contributions. The hotspots of k-cliques indicate research focuses, containing publications of small subdivision that builds upon all prior publications.

However, there exist several major challenges with existing heuristic and GNN approaches: 1) The number of possible query graph structures and subgraph counts both grow combinatorially with respect to the graph size Sloane (2014); Read & Wilson (1998), resulting in large approximation error Ribeiro et al. (2021). The count can have high dynamic range from zero to millions, making the task much harder than most graph regression tasks which only predict a single-digit number with a small upperbound. 2) The expressive power of commonly used message passing GNNs is limited by the Weisfeiler-Lehman (WL) test Leman & Weisfeiler (1968); Chen et al. (2020); Xu et al. (2018). Certain structures are not distinguishable with these GNNs, let alone counting them, resulting in the same count prediction for different queries. 3) Furthermore, most existing approximate heuristic and GNN methods only focus on estimating the total count of a query in the target graph Bressan et al. (2019); Liu et al. (2020); Chen et al. (2020), but not the distribution of occurrences of the patterns, as shown in Figure 1. Yet such distribution information is crucial in various applications Yin et al. (2019); Tsourakakis et al. (2017); Benson et al. (2016); Faust (2010); Holland & Leinhardt (1976).

**Proposed work**. To resolve the above challenges, we propose DeSCo, a GNN-based model that learns to predict both pattern counts and distribution on any target graph. The main idea of DeSCo is to leverage and organize local information of neighborhood patterns to predict query count and distribution in the entire target graph. DeSCo first uses *canonical partition* to decompose the target graph into small neighborhoods without missing and double-counting any patterns. The local information is then encoded using a GNN with *subgraph-based heterogeneous message passing*. Finally, we perform *gossip correction* to improve counting accuracy. Our contributions are three-fold.

**Canonical partition**. Firstly, we propose a novel divide-and-conquer scheme called *canonical partition* to decompose the problem into subgraph counting for individual neighborhoods. The canonical partition ensures that no pattern will be double counted or missed over all neighborhoods. The algorithm allows the model to make accurate predictions even with the high dynamic range of labels and enables subgraph count distribution prediction for the first time. Figure 1 demonstrates DeSCo's predictions on the query graph count distribution of a citation network. The count hotspots of different queries can indicate citation patterns of different scientific communities Gao & Lafferty (2017); Yang et al. (2015), which shed light on the research impact of works in this network.

**Subgraph-based heterogeneous message passing**. Secondly, we propose a general approach to enhance the expressive power of any MPGNNs by encoding the subgraph structure through heterogeneous message passing. The message type is determined by whether the edge presents in a certain subgraph, e.g., a triangle. We theoretically prove that its expressive power can exceed the upper bound of that of MPGNNs. We show that this architecture outperforms expressive GNNs, including GIN Xu et al. (2018) and ID-GNN You et al. (2021).

**Gossip correction**. We overcome the challenge of accurate count prediction by utilizing two inductive biases of the counting problem: homophily and antisymmetry. Real-world graphs share similar patterns among adjacent nodes, as shown in Figure 1. Furthermore, since canonical count depends

Figure 2: DeSCo Framework in 3 steps. **(a) Step 1. Canonical Partition**: Given *query* and *target*, decomposed *target* into multiple node-induced subgraphs, i.e., *canonical neighborhood*s, based on node indices. Each neighborhood contains a *canonical node* that has the greatest index in the neighborhood. **(b) Step 2. Neighborhood Counting**: Predict the *canonical count*s of each neighborhood via an expressive GNN, and assign the count of the neighborhood to the corresponding *canonical node*. Neighborhood counting is the local count of queries. **(c) Step 3. Gossip Correction**: Use GNN prediction results to estimate *canonical count*s on the *target* graph through learnable gates.

on node indices, there exists antisymmetry due to canonical partition. Therefore, we propose a *gossip correction* phase, featuring a learnable gate for propagation to leverage the inductive biases.

To demonstrate the effectiveness of DeSCo, we compare it against state-of-the-art exact and approximate heuristic methods for subgraph counting as well as recent GNN-based approaches Cordella et al. (2004); Bressan et al. (2019); Chen et al. (2020); Liu et al. (2020), in terms of both performance and runtime efficiency. Experiments show that DeSCo enables large-scale subgraph counting that was not possible for exact methods. Compared with heuristic and neural methods, DeSCo achieves more than two orders of magnitude improvement in the mean squared error of count prediction. To the best of our knowledge, it is also the first method to enable accurate count distribution prediction. Furthermore, the model excels in both accuracy and runtime efficiency for larger queries, with the highest percentage of valid predictions and up to two orders of magnitude speedup over heuristic methods. Our code is available at `https://anonymous.4open.science/r/DeSCo-6BD2`

## 2 RELATED WORKS

There has been an extensive line of work to solve the subgraph counting problem.

**Exact counting algorithms**. Exact methods generally count subgraphs by searching through all possible node combinations and finding the matching pattern. Early methods usually focus on improving the matching phase Wernicke & Rasche (2006); Cordella et al. (2004); Milo et al. (2002) Recent approaches emphasize pruning the search space and avoiding double counting Demeyer et al. (2013); Mawhirter et al. (2019); Shi et al. (2020); Mawhirter & Wu (2019). However, exact methods still scale poorly in terms of query size (often no more than five nodes) despite much efforts Pinar et al. (2017); Chen & Qian (2020).

**Approximate heuristic methods**. To further scale up the counting problem, approximate counting algorithms sample from the target graph to estimate pattern counts. Strategies like path sampling Wang et al. (2017); Jha et al. (2015), random walk Yang et al. (2018); Saha & Hasan (2015), substructure sampling Fu et al. (2020); Iyer et al. (2018), and color coding Bressan et al. (2021; 2018) are used to narrow the sample space and provides better error bound. However, large and rare queries are hard to find in the vast sample space, leading to large approximation error Bressan et al. (2019).

**GNN-based approaches**. Recently, GNNs have also been used to attempt subgraph counting. Liu et al. (2020) uses GNNs to embed the query and target graph, and predict subgraph counts via embeddings. Chen et al. (2020) theoretically analyzes the expressive power of GNNs for counting and proposes an expressive GNN architecture. Zhao et al. (2021) proposes an active learning scheme for the problem. Unfortunately, large target graphs have extremely complex structures and a high dynamic range of pattern count, so accurate prediction remains challenging.

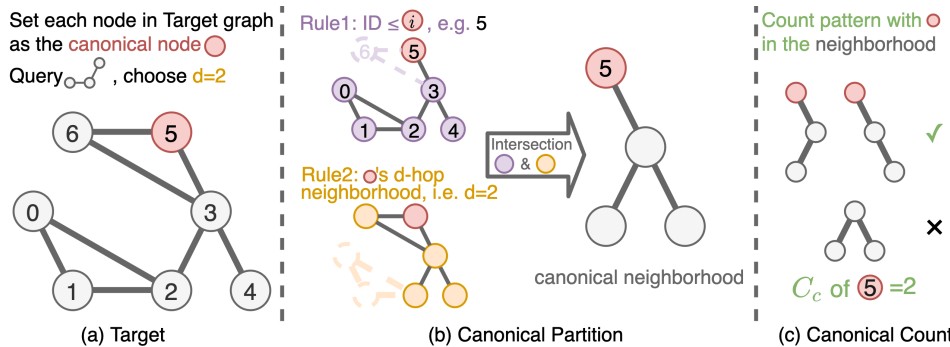

Figure 3: An example of canonical partition and canonical count. (a) Choose node 5 from the target graph as the *canonical node* (red circle). (b) *Canonical partition* generates the corresponding *canonical neighborhood* graph. It performs an ID-restricted breadth-first search to find the induced neighborhood that complies with both Rule1 and Rule2. (c) The corresponding *canonical count* is defined by the number of patterns containing the canonical node in the canonical neighborhood. DeSCo's *neighborhood counting* phase predicts the canonical count for each canonical neighborhood.

## 3 PRELIMINARY

Let $G_t = (V_t, E_t)$ be a large *target* graph with vertices $V_t$ and edges $E_t$. Let $G_q = (V_q, E_q)$ be the *query* graph of interest. The *subgraph counting problem* $\mathcal{C}(G_q, G_t)$ is to calculate the size of the *set of patterns* $\mathcal{P} = \{G_p | G_p \subseteq G_t\}$ in the target graph $G_t$ that are isomorphic to the query graph $G_q$, that is, $\exists$ bijection $f : V_p \mapsto V_q$ such that $(f(v), f(u)) \in E_q$ iff $(v, u) \in E_p$, denoted as $G_p \cong G_q$.

Subgraph counting includes induced and non-induced counting depending on whether the pattern $G_p$ is restricted to induced subgraph Ribeiro et al. (2021). A $G_p = (V_p, E_p)$ is induced subgraph of $G_t$ if $\forall e \in E_t \leftrightarrow e \in E_p$, denoted as $G_p \subseteq G_t$. Without loss of generality, we focus on the connected, induced subgraph counting problem, following modern mainstream graph processing frameworks Hagberg et al. (2008); Peixoto (2014) and real-world applications Wong et al. (2012); Milo et al. (2002). It is also possible to obtain non-induced occurrences from induced ones with a transformation Floderus et al. (2015). Our approach can easily support graphs with node features and edge directions. But to compare with heuristic methods that only support simple graphs, we use undirected graphs without node features to investigate the ability to capture graph topology.

## 4 DESCO FRAMEWORK

In this section, we introduce the pipeline of DeSCo. To perform subgraph counting, DeSCo first performs **canonical partition** to decompose the target graph to many canonical neighborhood graphs. Then, **neighborhood counting** uses the subgraph-based heterogeneous GNN to embed the query and neighborhood graphs and performs a regression task to predict the canonical count on each neighborhood. Finally, **gossip correction** propagates neighborhood count predictions over the target graph. We will first introduce the model objective before elaborating on each step.

### 4.1 CANONICAL COUNT OBJECTIVE

The canonical count is used as a local count prediction objective for the GNN and gossip correction, after decomposing the target into small neighborhoods without missing or double-counting patterns.

**Canonical node**. We use randomly assigned node indices on the target graph to break the symmetry of patterns. We assign the match of a k-node pattern to its *canonical node* based on the index. Formally, the *canonical node* $v_c$ is the node with the largest node index in the pattern: $v_c = \max_I V_p$.

The number of patterns that share the same canonical node is called the *canonical count* $\mathcal{C}_c$ on this node as shown in Figure 3 (c). Note how the match of a k-node pattern is only attributed to the canonical node, since the other k-1 nodes do not satisfy $v = \max_I V_p$. It also suggests that the node with a relatively larger node index will result in a larger canonical count. This characteristic will be utilized by gossip correction discussed in Section 4.4.

**Definition 4.1** (canonical count).

$$\mathcal{C}_c(G_q, G_t, v_c) = |\{G_p \subseteq G_t | G_p \cong G_q, v_c = \max_I V_p\}| \tag{1}$$

We prove the following Lemma in Appendix A.1.

**Lemma 4.1.** *The subgraph count of query in target equals the summation of the canonical count of query in target over all target nodes.*

$$\mathcal{C}(G_q, G_t) = \sum_{v_c \in V_t} \mathcal{C}_c(G_q, G_t, v_c) \tag{2}$$

Lemma 4.1 allows the decomposition of the counting problem into multiple neighborhood canonical counting objectives. We use the following canonical partition for the decomposition.

## 4.2 CANONICAL PARTITION

Based on Lemma 4.1, we define **canonical partition** that decomposes the target graph. Figure 3 (a), (b) shows an example of the canonical neighborhood, obtained by a partition function $\mathcal{P}$ defined as:

$$\mathcal{P}(G_t, v_c, d) = G_c, \mathrm{s.\,t.}\, G_c \subseteq G_t, V_c = \{v_i \in V_t | \mathcal{D}(G_t, v_i, v_c) \le d, v_i \le v_c\} \tag{3}$$

$\mathcal{D}(G_t, v_i, v_c)$ means the shortest distance between $v_i$ and $v_c$ on $G_t$. We further prove:

**Theorem 1.** *The subgraph count of query in target equals the summation of the canonical count of query in canonical neighborhoods over all target nodes. Canonical neighborhoods are acquired with partition $\mathcal{P}$, given any $d$ greater than the diameter of query.*

$$\mathcal{C}(G_q, G_t) = \sum_{v_c \in V_t} \mathcal{C}_c(G_q, \mathcal{P}(G_t, v_c, d), v_c), d \ge \max_{v_i, v_j \in V_q} \mathcal{D}(G_q, v_i, v_j) \tag{4}$$

**Definition 4.2** (Canonical partition). *Given target graph $G_t$, canonical partition iterates over all nodes $v$ of the target $G_t$ and partition it into a set of canonical neighborhoods $G_{v_c}$.*

$$G_t \mapsto \{G_{v_c} | G_{v_c} = \mathcal{P}(G_t, v_c, d), v_c \in V_t\} \tag{5}$$

Missing count or double counting are avoided with Theorem 1 and Definition 4.2, which is detailedly proven in Appendix A.2. In practice, we set $d$ as the maximum diameter of query graphs to meet the requirements of Theorem.1. See Appendix A.3 for implementation of $\mathcal{P}(G_t, v_c, d)$. Appendix A.3 shows that canonical partition reduces the complexity of the problem by a dozen orders of magnitude. After canonical partition, DeSCo uses a GNN to predict the canonical count for each decomposed neighborhood. This divide-and-conquer scheme not only greatly reduces the complexity of each GNN prediction, but also makes it possible to predict the count distribution over the entire graph.

## 4.3 NEIGHBORHOOD COUNT PREDICTION

After canonical partition, GNNs are used to predict the *canonical count* $C_c(G_q, G_{v_c}, v_c)$ on any canonical neighborhood $G_{v_c}$. The canonical neighborhood and the query are separately embedded using GNNs. The embeddings are passed to a multilayer perceptron to predict the canonical count.

To strengthen the expressive power of the GNN used in neighborhood count, we propose a general Subgraph-based Heterogeneous Message Passing (SHMP), which incorporates topological information through node and edge types. The canonical node of the neighborhood is treated as a special node type. SHMP further uses small subgraph structures to categorize edges into different edge types, and use different learnable weights for each edge type.

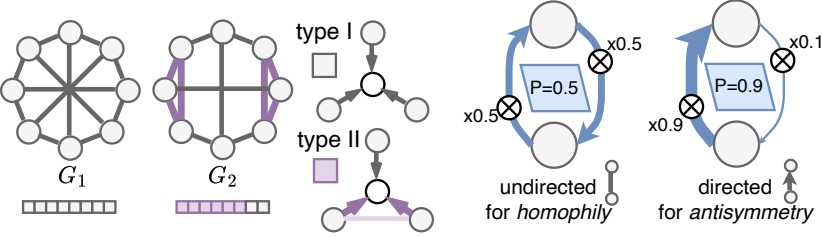

(a) SHMP for neighborhood counting     (b) Learnable gate for gossip correction

Figure 4: (a) Proposed SHMP. Embedded with regular MP, graphs $G_1$ and $G_2$ are indistinguishable. While embedded with SHMP, $G_2$ is successfully distinguished with six type II node embeddings, demonstrating better expressive power of SHMP. (b) Proposed learnable gates in the gossip model balance the influence of *homophily* and *antisymmetry* by controlling message directions.

**Definition 4.3** (subgraph-based heterogeneous message passing). *The SHMP uses the following equation 6 to compute each node's representation at layer $k$. Here $\phi_h^k$ denotes the message function of the h-th edge type. $N_h(i)$ denotes nodes that connect to node i with the h-th edge type.* AGG *and* AGG′ *are the permutation invariant aggregation function such as sum, mean, or max.*

$$\mathbf{x}_i^{(k)} = \gamma^{(k)}\left(\mathbf{x}_i^{(k-1)}, \text{AGG}'_{h\in H}\left(\text{AGG}_{j\in N_h(i)}\left(\phi_h^{(k)}(\mathbf{x}_i^{(k-1)}, \mathbf{x}_j^{(k-1)}, \mathbf{e}_{j,i})\right)\right)\right) \tag{6}$$

Note that MP defined by major GNN frameworks Fey & Lenssen (2019); Wang et al. (2019) is just a special case of SHMP if only one edge type is derived with the subgraph structure. We theoretically prove that SHMP can exceed the upper bound of MP in terms of expressiveness in Appendix B.1.

For example, Figure 4(a) demonstrates that triangle-based heterogeneous message passing has better expressive power. Regular MPGNNs fail to distinguish different d-regular graphs $G_1$ and $G_2$ because of their identical type I messages and embeddings, which is a common problem of MPGNNs You et al. (2021). SHMP, however, can discriminate the two graphs by giving different embeddings. The edges are first categorized into two edge types based on whether they exist in any triangles (edges are colored purple if they exist in any triangles). Since no triangles exist in $G_2$, all of its nodes still receive type I messages. While some nodes of $G_1$ now receive type II messages with two purple messages and one gray message in each layer. As a result, the model acquires not only the adjacency information between the message sender and receiver, but also information among their neighbors. Such subgraph structural information improves expressiveness by incorporating high-order information in both the query and the target.

## 4.4 GOSSIP CORRECTION

Given the count predictions $\hat{C}_c$ output by the GNN, DeSCo uses **gossip correction** to improve the prediction quality, enforcing different homophily and antisymmetry inductive biases for different queries. Gossip correction uses another GNN to model the error of neighborhood count. It uses the predicted $\hat{C}_c$ as input, and the canonical counts $C_c$ as the supervision for corresponding nodes in the target graph.

**Motivation**. Two different inductive biases are used to improve the accuracy. 1) *Homophily*. Since the neighborhoods of adjacent nodes share much common graph structure, they tend to have similar canonical counts as shown in Figure1. This is called the *homophily* of canonical counts. 2) *Antisymmetry*. As mentioned in Section4.1, for nodes with similar neighborhood structures, the one with a larger node index has a larger canonical count, resulting in *antisymmetry* of canonical counts. See the example target graph in Figure 2, which satisfies the inductive biases. Appendix C further shows that *homophily* and *antisymmetry* are mutually exclusive for different queries, which corresponds to our gossip model design in Figure 4(b).

**Gossip correction with learnable gates**. As shown in Figure 4(b), The proposed gossip model multiplies a learnable gate $P$ for the message sent from the node with the smaller index, and $1 - P$ from the reversed one. $P$ is learned from the query embedding. For different queries, $P$ ranges from 0 to 1 to balance the influence of *homophily* and *antisymmetry*. When $P \to 0.5$, messages from the smaller indexed node and the reversed one are weighed equally. So it simulates undirected message

passing that stress *homophily* by taking the average of adjacent node values. When the gate value moves away from 0.5, the message from a certain end of the edge is strengthened. For example, when $P \to 1$, the node values only accumulate from nodes with smaller indices to nodes with larger ones. So that it simulates directed message passing that stress *antisymmetry* of the transitive partial order of node indices.

The messages of MPGNNs are multiplied with $g_{ji}$ on both edge directions. With the learnable gates, the model now better utilizes the mutually exclusive inductive biases for better error correction.

$$\mathbf{x}_i^{(k)} = \gamma^{(k)} \left( \mathbf{x}_i^{(k-1)}, \text{AGG}_{j \in N(i)} g_{ji} \cdot \phi^{(k)} \left( \mathbf{x}_i^{(k-1)}, \mathbf{x}_j^{(k-1)}, \mathbf{e}_{j,i} \right) \right), g_{ji} = \begin{cases} P & v_j \leq v_i \\ 1 - P & v_j > v_i \end{cases} \quad (7)$$

**Final count prediction**. The gossip-corrected neighborhood count is a more accurate estimation of the canonical count and distribution. The summation of the (corrected) neighborhood count is the unbiased estimation of subgraph count on the whole target graph as Theorem 1 states.

## 5 EXPERIMENTS

We compare the performance of DeSCo with state-of-the-art approximate heuristic and neural subgraph counting methods. The runtime advantage is also demonstrated with popular exact methods. Extensive ablation studies further show the benefit of each component of DeSCo.

### 5.1 EXPERIMENTAL SETUP

| Dataset | #graphs | Avg. #nodes | Avg. #edges |
|---------|---------|-------------|-------------|
| MUTAG | 188 | 17.93 | 19.79 |
| COX2 | 467 | 41.22 | 43.45 |
| ENZYMES | 600 | 32.63 | 62.14 |
| SYNTHETIC | 6400 | 41.58 | 158.81 |
| CITESEER | 1 | 3.3K | 4.5K |
| CORA | 1 | 2.7K | 5.4K |

Table 1: Graph statistics of datasets used in experiments.

| Dataset | CiteSeer | | | Cora | | |
|---------|----------|---|---|------|---|---|
| Query-Size | 3 | 4 | 5 | 3 | 4 | 5 |
| LRP | GPU out of memory | | | overflow | | |
| DIAMNet | 1.110 | 1.282 | 1.101 | 1.074 | 1.108 | 1.037 |
| DeSCo | **0.006** | **0.133** | **0.125** | **0.302** | **0.249** | **0.676** |

Table 2: Normalized MSE performance of neural methods on large targets with standard queries.

**Datasets**. We use real-world datasets from various domains as the target graphs, including chemistry(MUTAG Debnath et al. (1991), COX2 Rossi & Ahmed (2015)), biology(ENZYMES Borgwardt et al. (2005)), and citation networks(CiteSeer Giles et al. (1998), Cora McCallum et al. (2000)). We also generate a large synthetic dataset with mixed graph generators Holme & Kim (2002); Albert & Barabási (2000); Watts & Strogatz (1998); Erdős et al. (1960) to cover diverse graph characteristics. All the datasets are treated as undirected graphs without node or edge features in alignment with the setting of the approximate heuristic method Bressan et al. (2019). The *standard query* graphs include all non-isomorphic, connected, undirected graphs with node size $3 - 5$. The ground truth total counts and canonical counts of these queries are generated with the exact counting method on all the target graphs from the above datasets.

**Pretraining**. To perform subgraph counting on any target graph, we first pre-train all the neural methods with the target-query pairs from the synthetic dataset and the standard queries of size $3 - 5$. The neural baselines are trained and tested with the total subgraph count objective. After pretraining, DeSCo can be evaluated by predicting the total count in alignment with the baselines. Unless specified, the trained models are directly tested with standard queries and the targets from unseen, real-world datasets. Therefore, in our evaluation, DeSCo only needs to be trained once from scratch across common datasets and tasks.

**Baselines**. For neural-based approaches, we adopt state-of-the-art subgraph counting GNNs, LRP Chen et al. (2020) and DIAMNet Liu et al. (2020). For the approximate heuristic counting method, we choose the state-of-the-art color-based sampling method MOTIVO Bressan et al. (2019). It uses color coding to greatly narrow the sample space with efficient c++ implementation. For exact counting methods, we consider VF2 Cordella et al. (2004) and IMSM Sun & Luo (2020). VF2 is widely used in major graph processing frameworks Hagberg et al. (2008); Peixoto (2014).

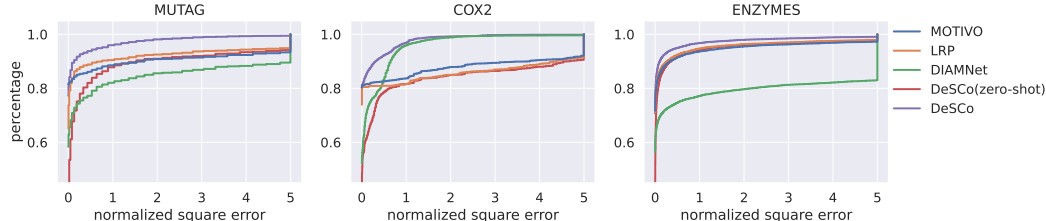

Figure 5: The accumulative distributions of normalized square error of large query-target pairs. The x-axis is clipped at 5. Given any square error tolerance bound (x-axis), DeSCo has the highest percentage of predictions that meet the bound (y-axis). DeSCo(zero-shot) generalizes to unseen queries with competitive performance over specifically trained baselines.

IMSM is a common framework that combines optimizations from multiple exact counting methods He & Singh (2008); Bonnici et al. (2013); Bhattarai et al. (2019); Han et al. (2019). Refer to Appendix D.3 and E for the configuration details of the baselines.

**Evaluation metric**. We follow the previous works Chen et al. (2020); Liu et al. (2020) and use mean square error (MSE) of total subgraph count prediction as our evaluation metric. The MSE values of each query size are normalized by dividing the variance of the ground truth counts.

## 5.2 NEURAL COUNTING

Table 3 summarizes the normalized MSE for predicting the subgraph count of twenty-nine standard query graphs on datasets consisting of many target graphs. With canonical partition, neighborhood counting, and gossip correction, DeSCo demonstrates $437\times$ improvements against the best baseline on average. This demonstrates that neural subgraph counting is truly reliable in real-world problems, even for smaller queries of size $3-5$. The relative count error under the q-error metric is also discussed in Appendix G.1. DeSCo also highlights the accurate count distribution prediction for the first time. The distribution prediction realizes $0.23$ normalized MSE as discussed in Appendix F.

| Dataset | MUTAG | | | COX2 | | | ENZYMES | | |
|---|---|---|---|---|---|---|---|---|---|
| Query-Size | 3 | 4 | 5 | 3 | 4 | 5 | 3 | 4 | 5 |
| MOTIVO | 1.1E+2 | 7.5E+2 | 4.8E+3 | 3.4E+2 | 2.9E+3 | 3.2E+4 | 1.3E+2 | 6.0E+2 | 3.6E+3 |
| LRP | 1.6E+0 | 1.1E+0 | 1.0E+0 | 1.9E+0 | 1.3E+0 | 1.1E+0 | 2.0E+0 | 1.3E+0 | 1.1E+0 |
| DIAMNet | 1.7E-1 | 1.1E-1 | 3.5E-1 | 3.0E-1 | 2.4E-1 | 5.0E-1 | 7.2E-1 | 5.6E-1 | 8.9E-1 |
| DeSCo | **7.3E-5** | **5.2E-4** | **1.1E-2** | **2.3E-5** | **9.5E-5** | **7.2E-3** | **1.1E-3** | **2.0E-3** | **1.0E-2** |

Table 3: Normalized MSE of approximate heuristic and neural methods on subgraph counting of twenty-nine standard queries.

## 5.3 SCALIBILITY

**Generalization**. Obtaining ground truth for large queries and targets via exact counting is extremely expensive and can take months, so we only test scalable queries and targets with the following setting in Section 5.3. Here we demonstrate that with minimal pre-training, the model is able to generalize and make reliable predictions for larger queries and targets.

**Large queries**. For each query size between 6 to 13, we select two queries that frequently appear in ENZYMES. A smaller synthetic dataset with 2048 graphs is used to generate the ground truth for these sixteen queries. All the models are pre-trained with standard queries. All the models, except for DeSCo(zero-shot), are fine-tuned with larger queries on the small synthetic dataset. DeSCo(zero-shot) is used to show the generalization power of DeSCo for unseen queries. The distributions of the square error of each query-target pair are shown in Figure 5. The square errors are normalized with the variance of all ground truth counts.

**Large target**. We also test the models on large target graphs shown in Table 2. The maximum ground truth count of standard queries goes up to $3.8 \times 10^6$ and $3.3 \times 10^7$ on CiteSeer and Cora, thus a hard task. When directly tested, LRP and DIAMNet overflow normal float precision. So we tune all the models on one graph and test on the other to simulate transfer learning between graphs from the same domain. LRP either exceeds 32GB GPU memory in tuning, or predicts infinite caused by overflow.

## 5.4 ABLATION STUDY

We explore the effectiveness of each component of DeSCo through the ablation study by removing each component.

**Ablation of canonical partition**. We remove the canonical partition of DeSCo and train it with the objective of subgraph count on the whole target, the same as other neural baselines. This demonstrates the effectiveness of the divide-and-conquer scheme of canonical partition.

| Dataset | MUTAG | | | COX2 | | | ENZYMES | | |
|---------|-------|-------|-------|-------|-------|-------|---------|---------|---------|
| Query-Size | 3 | 4 | 5 | 3 | 4 | 5 | 3 | 4 | 5 |
| w/o $\mathcal{P}$ | 1.8E-2 | 1.0E-2 | 6.4E-2 | 2.1E-2 | 1.5E-2 | 2.5E-2 | 7.3E-1 | 1.7E+0 | 3.9E+0 |
| w $\mathcal{P}$ | **4.2E-5** | **1.4E-4** | **7.8E-3** | **1.8E-5** | **2.3E-5** | **1.5E-3** | **9.6E-4** | **3.0E-3** | **1.2E-2** |

Table 4: Normalized MSE performance with or without canonical partition. Since gossip correction relies on the output of neighborhoods, it's also removed for both for a fair comparison.

**Ablation of subgraph-based heterogeneous message passing**. We use our proposed SHMP to improve the performance of GraphSAGE by transforming its standard message passing to heterogeneous message passing. We use triangle as the subgraph to categorize heterogeneous edges as shown in Figure 4(a). Note how SHMP outperforms expressive GNNs, including GIN and ID-GNN.

| Dataset | MUTAG | | | COX2 | | | ENZYMES | | |
|---------|-------|-------|-------|-------|-------|-------|---------|---------|---------|
| Query-Size | 3 | 4 | 5 | 3 | 4 | 5 | 3 | 4 | 5 |
| GCN | 1.7E+1 | 7.5E+0 | 8.2E-1 | 1.6E+01 | 6.6E+0 | 5.1E-1 | 8.7E-1 | 3.6E-1 | 7.7E-1 |
| SAGE | 9.2E-4 | 4.8E-3 | 1.5E-2 | 2.6E-5 | 3.5E-4 | 6.0E-4 | 3.0E-2 | 9.6E-2 | 2.4E-1 |
| GIN | 1.4E-4 | 3.0E-3 | 3.0E-2 | 6.6E-5 | 4.8E-4 | 1.0E-2 | 4.0E-2 | 1.4E-1 | 3.0E-1 |
| ID-GNN | 4.2E-5 | 3.7E-4 | **7.4E-3** | **1.6E-5** | 1.4E-4 | 2.2E-3 | **8.8E-4** | 5.5E-3 | 2.0E-2 |
| SAGE+SHMP | **4.2E-5** | **1.4E-4** | 7.8E-3 | 1.8E-5 | **2.3E-5** | **1.5E-3** | 9.6E-4 | **3.0E-3** | **1.2E-2** |

Table 5: Normalized MSE performance with different GNN models for neighborhood counting.

**Ablation of gossip correction**. The normalized MSE of direct summation of neighborhood counts and the summation after gossip correction are compared to show the effectiveness of gossip correction. We use ENZYMES and Cora to do the ablation study since the error of other graphs is already very low with only canonical partition and neighborhood counting.

| Dataset | ENZYMES | | | Cora | | |
|---------|---------|---------|---------|---------|---------|---------|
| Query-Size | 3 | 4 | 5 | 3 | 4 | 5 |
| w/o gossip | **9.6E-4** | 3.0E-3 | 1.2E-2 | 3.0E-1 | 2.5E-1 | 6.8E-1 |
| w gossip | 1.1E-3 | **2.0E-3** | **1.0E-2** | **2.9E-1** | **1.8E-1** | **6.3E-1** |

Table 6: The normalized MSE performance with and without gossip correction.

## 5.5 RUNTIME COMPARISON

DeSCo benefits from polynomial time complexity: $O(E_t^{3/2} + V_t \times (\bar{V}_n + \bar{E}_n)) + O(E_q^{3/2} + V_q)$. In contrast, both the exact methods VF2, IMSM, and the approximate heuristic method MOTIVO suffer from exponentially growing runtime with regard to the query size in practice. Experiments show that DeSCo scales to much larger queries than the exact methods and achieves up to $120\times$ speedup over the approximate heuristic method for large queries. The detailed runtime comparison is discussed in Appendix E.

## 6 CONCLUSION

We propose DeSCo, a neural network based framework for scalable subgraph counting. With canonical partition, subgraph-based heterogeneous message passing, and gossip correction, DeSCo accurately predicts counts for both large queries and targets. It demonstrates two magnitudes of improvements in mean square error and runtime efficiency. It additionally provides the important locational distribution of patterns that previous works cannot.

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

# A CANONICAL PARTITION

## A.1 PROOF OF LEMMA 4.1

*Proof.* Following the notions from Section 3, given a query graph $G_q$ and a target graph $G_t$, the node-induced count is defined as the number of $G_t$'s node-induced subgraph, pattern, $G_p$ that is isomorphic to $G_q$. We denote the set of all $G_p$ as $\mathbb{M}$.

$$\mathbb{M} = \{G_p \subseteq G_t | G_p \cong G_q\} \tag{8}$$
$$\mathcal{C}(G_q, G_t) = |\mathbb{M}| \tag{9}$$

Assuming that $G_q$ has k nodes. Then, under the node-induced definition, given $G_t$, we can use the k-node set $V_p = \{v | v \in G_p\}$ of $G_p$ to represent the pattern.

We can decompose the set of all patterns $\mathbb{M}$ into many subsets $\mathbb{M}_c$ based on the maximum node index of each $G_p \in \mathbb{M}$.

$$\mathbb{M}_c = \{G_p \subseteq G_t | G_p \cong G_q, \max_I V_p = c\} \tag{10}$$

This maximum-index decomposition is exclusive and complete: every $G_p$ has a single corresponding maximum node index. So we have the following properties:

$$\forall c \neq j, \mathbb{M}_c \cap \mathbb{M}_j = \varnothing \tag{11}$$

$$\mathbb{M} = \bigcup_{c=0}^{|V|-1} \mathbb{M}_c \tag{12}$$

Thus, the node-induced count in Equation 9 can be rewritten using the inclusion-exclusion principle.

$$
\begin{aligned}
\mathcal{C}(G_q, G_t) &= \left| \bigcup_{c=0}^{|V|-1} \mathbb{M}_c \right| \\
&= \sum_{c=0}^{|V|-1} |\mathbb{M}_c| + \sum_{k=1}^{|V|-1} (-1)^k \left( \sum_{0 \leq i_0 \leq \cdots i_k \leq |V|-1} |\mathbb{M}_{i_0} \cap \cdots \cap \mathbb{M}_{i_k}| \right) \\
&= \sum_{c=0}^{|V|-1} |\mathbb{M}_c|
\end{aligned}
\tag{13}
$$

According to the definition of canonical count in Equation 1, $\mathcal{C}_c(G_q, G_t, v_c) = |\mathbb{M}_c|$. Thus, Lemma 4.1 is proven with Equation 13.

$\square$

## A.2 PROOF OF THEOREM 1

*Proof.* By the definition of $\mathbb{M}_c$ in Equation 10, we have a corollary.

**Corollary 1.1.** *Denote $v_c$'s index as c, $\mathcal{D}$ as the shortest path length between two nodes. Any graph in $\mathbb{M}_c$ has node $v_c$ and has the same graph-level property with $G_q$, e.g., diameter.*

$$\forall G_p \in \mathbb{M}_c, v_c \in V_p, \max_{v_i, v_j \in V_p} \mathcal{D}(G_p, v_i, v_j) = \max_{v_i, v_j \in V_q} \mathcal{D}(G_q, v_i, v_j) \tag{14}$$

The distance between $v_c$ and any nodes of $G_p$ in $\mathbb{M}_c$ is bounded by $\max_{v_i,v_j \in V_q} \mathcal{D}(G_q, v_i, v_j)$ as shown in corollary 1.1. So we can further know that graphs in $\mathbb{M}_c$ are node-induced subgraphs of $v_c$'s d-hop ego-graph.

$$\forall G_p \in \mathbb{M}_c, \exists G_{d-ego} \subseteq G_t, V_{d-ego} = \{v_i \in V_t | \mathcal{D}(G_t, v_i, v_c) \leq d\} \,\text{s.t.}\, G_p \cong G_{d-ego} \qquad (15)$$

Given Equation 10, it is also clear that all graphs in $\mathbb{M}_c$ have smaller node indices than $c$.

$$\forall G_p \in \mathbb{M}_c, \exists G_{small} \subseteq G_t, V_{small} = \{v_i \in V_t | I_i \leq I_c\} \,\text{s.t.}\, G_p \cong G_{small} \qquad (16)$$

With Equation 15 and 16, we know that all the graphs in $\mathbb{M}_c$ are subgraphs of $\mathcal{P}(G_t, v_c, d)$ defined in Equation 3. Thus, with respect to Equation 8, we can redefine $\mathbb{M}_c$ as follows.

$$\mathbb{M}_c = \{G_p \subseteq \mathcal{P}(G_t, v_c, d) | G_p \cong G_q, \max_{V_p} I = c\} \qquad (17)$$

Combining Equation 13 with Equation 17, Theorem 1 is proven. $\qquad\qquad\qquad\square$

## A.3 IMPLEMENTATION OF CANONICAL PARTITION

---
**Algorithm 1** Index-restricted breadth-first search

---
$V_c \leftarrow \{v_c\}, V_{front} \leftarrow \{v_c\}$
**while** $depth < d$ **do**
$\quad V_{add} \leftarrow \{v | v \in \bigcup_{v_i \in V_{front}} \{v_j | (v_i, v_j) \in E_t\}, v \leq v_c\}$
$\quad V_{front} \leftarrow V_{add} \setminus V_c$
$\quad V_c \leftarrow V_c \cup V_{front}$
**end while**
$G_c \leftarrow (V_c, E_c) \,\text{s.t.}\, G_c \subseteq G_t$

---

The canonical partition is implemented using an index-restricted breadth-first search (BFS). Compared with regular BFS, it restricts the frontier nodes to have smaller indices than that of the canonical node. The time complexity of canonical partition equals to the BFS on each neighborhood $G_n = (V_n, E_n)$, which is $\sum O(V_n + E_n) = O(V_t \times (\bar{V}_n + \bar{E}_n))$.

## A.4 COMPLEXITY BENEFIT OF CANONICAL PARTITION

We discuss the computational benefit of the canonical partition method in this section.

**Search space reduction**. Canonical partition uses the divide-and-conquer scheme to bring about drastic search space reduction. We denote the complexity of searching and counting all subgraphs on size $V_t$ target graph as $S(V_t)$. Canonical partition divides the original problem into subproblems with the total search space of $\sum_{i \in V_t} S(V_{n_i})$, where $V_{n_i}$ stands for the size of canonical neighborhoods. Thanks to the sparse nature of real-world graphs, $V_{n_i}$s are generally small, even for huge target graphs. So with canonical partition, the search space is drastically reduced.

We conduct experiments on real-world graphs to show how canonical partition fundamentally reduces the search space. Figure 6 shows the computational complexity with different assumptions on the form of S. VF2 Cordella et al. (2004) claims that the asymptotic complexity for the problem ranges from $O(V^2)$ to $O(V! \times V)$ in best and worst cases. Under such assumptions of S, the average worst-case complexity is reduced by a factor of $1/10^{70}$ with canonical partition, while the average best-case complexity stays in the same magnitude. Empirically, we observe exponential runtime growth of the subgraph counting problem. Thus, under the assumption that $S(V) = 2^V$, the average complexity is also reduced drastically by a factor of $1/10^{11}$ with canonical partition.

**Redundant match elimination**. Canonical partition, along with the canonical count definition, eliminates the redundant automorphic match of the query graph. Previous works Mawhirter &

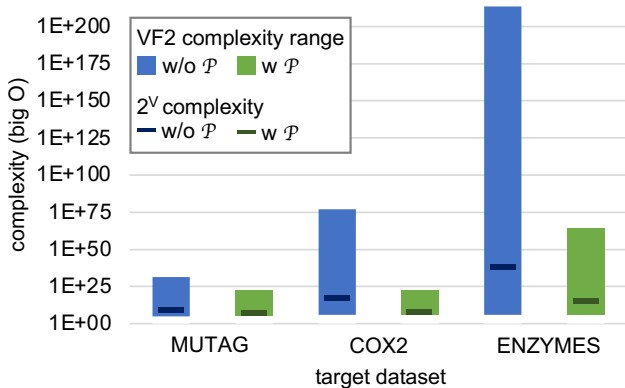

Figure 6: The complexity of subgraph counting with and without canonical partition on different target datasets. The complexity for the VF2 exact subgraph counting method is $O(V^2)$ to $O(V! \times V)$. The $O(2^V)$ complexity estimates the empirically observed average complexity.

Wu (2019); Shi et al. (2020) have shown that the automorphism of the query graph can cause a large amount of redundant count. For example, the triangle query graph $G_q$ has three symmetric nodes. We denote the triangle pattern as $G_p \subseteq G_t$ and the bijection $\mathbb{R}^3 \mapsto \mathbb{R}^3$ as $f : (v_{p_0}, v_{p_1}, v_{p_2}) \mapsto (v_{q_0}, v_{q_1}, v_{q_2})$. For the same pattern, there exist six bijections $\{f : (v_{p_0}, v_{p_1}, v_{p_2}) \mapsto (v_{q_i}, v_{q_j}, v_{q_k}) | (i, j, k) \in \mathrm{Perm}(1, 2, 3)\}$ where $\mathrm{Perm}(x, y, z)$ denotes all $3!$ permutations of $(x, y, z)$.

Canonical partition eliminates such redundant bijections by adding asymmetry, the canonical node. As discussed in Equation 1, by attributing the count to only one canonical node, the bijection $f_c$ can be rewritten as a $\mathbb{R}^3 \mapsto \mathbb{R}$ function, $f_c : (v_{p_0}, v_{p_1}, v_{p_2}) \mapsto \max_I(v_{q_0}, v_{q_1}, v_{q_2})$. It means that each query corresponds to only one bijection instead of six, thus preventing double counting and reducing the computational complexity.

**Reduction in the dynamic range of labels**. Canonical partition also reduces the dynamic range of the subgraph count labels, which makes the regression task easier for the neural network as discussed in Section 1. The detailed statistics of the dynamic range are shown in Appendix D.2. The canonical partition reduces the dynamic range of labels to $1/3$ on average in Figure 9.

## B  EXPRESSIVE POWER OF SHMP

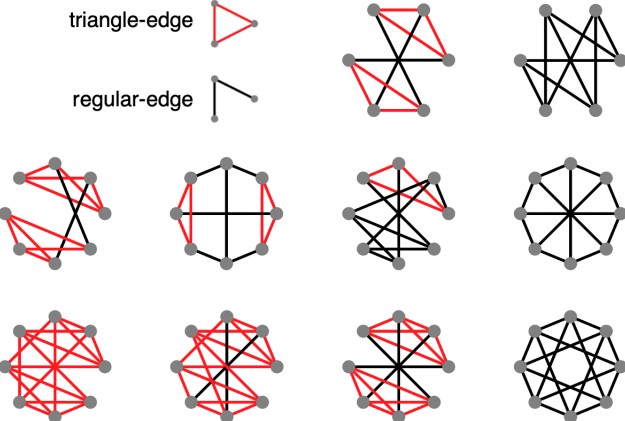

Figure 7: Examples of 1-SHMP distinguishable graphs. Any graph pair of each row cannot be distinguished by the 1-WL test. While with one layer of triangle-based SHMP, the histogram of the triangle-edge can distinguish all these graph pairs.

## B.1 THEORETICAL COMPARISON WITH REGULAR MESSAGE PASSING

Previous work Xu et al. (2018) has shown that the expressive power of existing message passing GNNs is upper-bounded by the 1-WL test, and such bound can be achieved with the Graph Isomorphism Network (GIN). We prove the expressive power of SHMP with the following Lemma.

**Lemma B.1.** *The SHMP version of GIN has stronger expressive power than the 1-WL test.*

By setting $\forall \phi_h^k = \phi^k$ and AGG$'$ = AGG, SHMP from Equation 6 becomes an instance of GIN, which proves that SHMP-GIN is at least as expressive as GIN or the 1-WL test. The examples in Figure 7 and Table 7 further prove that one layer of triangle-based SHMP-GIN can distinguish certain graphs that the 1-WL test cannot. Thus, SHMP-GIN has stronger expressive power than the 1-WL test, exceeding the upper bound of regular message passing neural networks.

## B.2 EXPERIMENTS ON REGULAR GRAPHS

To further illustrate the expressive power of SHMP, we show the number of graph pairs that are WL indistinguishable but SHMP distinguishable in Table 7. We collect all the connected, d-regular graphs of sizes six to twelve from the House of Graphs Brinkmann et al. (2013). Among these 157 graphs, 654 pairs of graphs are indistinguishable by the 1-WL test, even with infinite iterations. In comparison, only 208 pairs are indistinguishable by the triangle-based SHMP with a single layer. So 68% of typical fail cases of the 1-WL test are easily solved with SHMP. Some examples are shown in Figure 7.

| Graph Size | 6 | 7 | 8 | 9 | 10 | 11 | 12 |
|---|---|---|---|---|---|---|---|
| Number of Graphs | 5 | 4 | 15 | 10 | 30 | 5 | 88 |
| Number of Graph Pairs | 10 | 6 | 105 | 45 | 435 | 10 | 3828 |
| WL Indistinguishable | 1 | 1 | 19 | 13 | 64 | 1 | 555 |
| SHMP Indistinguishable | 0 | 1 | 4 | 4 | 26 | 0 | 173 |
| Error Reduction | 100.0% | 0.0% | 78.9% | 69.2% | 59.4% | 100.0% | 68.8% |

Table 7: The number of indistinguishable d-regular graph pairs for the WL-test and SHMP.

## B.3 DISCUSSION ON SUBSTRUCTURE ENHANCED GNNS

Previous substructure enhanced GNNs Morris et al. (2019); Nikolentzos et al. (2020) focus on the idea of high-order abstractions of the graph. However, the direct instantiation of all high-order substructures poses significant runtime overhead, which is unfriendly for the large-scale subgraph counting problem. For example, Morris et al. (2019) has to add k-combinatorially many nodes to represent the corresponding k-order substructure. This results in massive memory overhead and heavy message passing computation. Though both of them use the three-node substructure information, experiments show that the five-layer DeSCo is 3.51× faster than the five-layer 1-2-3-GNN Morris et al. (2019) when embedding the same COX2 dataset. In contrast, DeSCo's subgraph-based heterogeneous message passing (SHMP) focuses on the idea of distinguishing different local graph structures. By categorizing the messages on the original graph, DeSCo efficiently uses the same amount of message passing computation as traditional MPGNNs, while providing better expressive power.

## C HOMOPHILY AND ANTISYMMETRY ANALYSIS

**Example and observation**. The *homophily* and the *antisymmetry* are two important inductive biases for the canonical count. The target graph in Figure 2 serves as a vivid example. The numbers in the green square indicate the canonical count value of each node. On the one hand, note that the adjacent nodes 3, 5, and 6 have the same count value of 2. Adjacent nodes 0, 1, and 2 also have the same value, 0. This homophily inductive bias suggests that taking an average of the adjacent node values can reduce the prediction error of individual nodes. On the other hand, though node 1 and node 5 have similar neighborhood graph structures, node 5 with a larger node index has a higher canonical

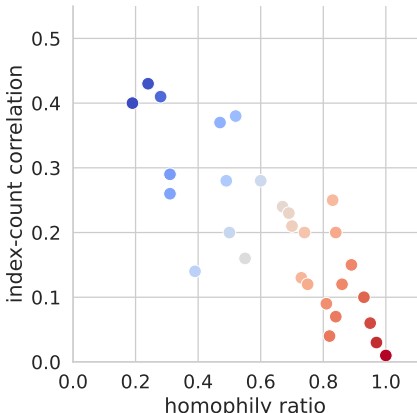

Figure 8: The quantification of *homophily* and *antisymmetry* of standard queries on the ENZYMES target graph. The extent of *homophily* and the *antisymmetry* are measured by the homophily ratio and the index-count correlation, respectively. The color corresponds to the x-coordinate (*homophily*) minus the y-coordinate (*antisymmetry*). Note how different queries emphasize one of the two inductive biases.

count value. It corresponds to the definition of canonical count as discussed in Section 4.1. This antisymmetry inductive bias suggests that the embedding phase for two structurally similar nodes with different node indices should also be different.

**Quantization**. We quantify the *homophily* and the *antisymmetry* inductive biases. For homophily, we treat the canonical count as the node label and use the homophily ratio from Zhu et al. (2021) to quantify how similar the count is between adjacent nodes. The homophily ratio ranges from 0 to 1. The higher the homophily ratio is, the more similar the labels will be between adjacent nodes. For antisymmetry, we use the Pearson correlation coefficient $r$ Benesty et al. (2009) between the node index and its canonical count as the quantification metric. We quantify the different *homophily* and *antisymmetry* for different queries on the ENZYMES target graph.

**Key insight**. As shown in Figure 8, the key insight is that *homophily* and the *antisymmetry* generally have negative correlation $r = -0.82$. So the emphasis on one should suppress the other. Based on such observation, we design the gossip correction model with learnable gates to imitate the mutually exclusive relation between the two inductive biases for different queries. As shown in Figure 4(b), the proposed learnable gate balances the influence of *homophily* and *antisymmetry* by controlling the direction of message passing. The gate value is trained to adapt for different queries to imitate different extents of *homophily* and *antisymmetry*.

# D EXPERIMENTAL SETUP

## D.1 SYNTHETIC DATASET

The DeSCo can be pre-trained once and be directly applied to any targets. So we generate a synthetic dataset with 6400 graphs, along with the ground truth count of all twenty-nine *standard queries* (queries of sizes 3, 4, 5).

The synthetic dataset generates each graph with a generator from the generator pool. The pool consists of four different graph generators: the Erdős-Rényi (ER) model Erdős et al. (1960), the Watts-Strogatz (WS) model Watts & Strogatz (1998), the Extended Barabási-Albert (Ext-BA) model Albert & Barabási (2000), and the Power Law (PL) cluster model Holme & Kim (2002). The expected graph size $n$ of each generator is uniformly chosen from 5 to 50. An additional 2% PL model is used to generate very large graphs with 50 to 250 nodes.

To generate an n-node graph, for the ER model, the edges are added with Beta probability distribution $p \sim 0.8\text{Beta}(1.3, 1.3n/\log_2 n - 1.3)$. For the WS model, each node connects to $K \sim n\text{Beta}(1.3, 1.3n/\log_2 n - 1.3)$ neighborhoods, with a rewiring probability $p \sim \text{Beta}(2, 2)$.

For the Ext-BA model, the edges are attached to each node with uniform probability distribution $m \sim \mathrm{U}(1, 2\log_2 n)$. The probability of adding edges and rewiring both conform to the clapped exponential distribution $p \sim \max(0.2, \mathrm{E}(20))$. For the PL model, the edges are attached to each node with uniform distribution $m \sim \mathrm{U}(1, 2\log_2 n)$. The triangle is added with $p \sim \mathrm{U}(0, 0.5)$.

## D.2 QUERY GRAPHS

Figure 9 shows all twenty-nine *standard queries* discussed in Section 5.1. They form the complete set of all non-isomorphic, connected, undirected graphs with three to five nodes. Figure 10 shows all sixteen *large query graphs* discussed in Section 5.3. They are frequent subgraphs with six to thirteen nodes in the ENZYMES dataset.

The figures also show the dynamic range (DR) of the ground truth count of these queries on the target graphs from the ENZYMES dataset. The canonical counts' dynamic range on the corresponding neighborhoods is also shown. Note how *canonical partition* reduces the dynamic range of the regression task for GNNs.

| $G_q$ | DR $G_t$ | DR $G_c$ | $G_q$ | DR $G_t$ | DR $G_c$ | $G_q$ | DR $G_t$ | DR $G_c$ |
|---|---|---|---|---|---|---|---|---|
| | 339 | 33 | | 1773 | 288 | | 37 | 15 |
| | 61 | 12 | | 195 | 48 | | 37 | 19 |
| | 282 | 48 | | 391 | 71 | | 19 | 12 |
| | 778 | 107 | | 499 | 121 | | 62 | 25 |
| | 244 | 42 | | 386 | 86 | | 23 | 13 |
| | 67 | 15 | | 44 | 13 | | 24 | 14 |
| | 67 | 17 | | 115 | 28 | | 9 | 5 |
| | 16 | 5 | | 156 | 51 | | 7 | 3 |
| | 203 | 80 | | 52 | 16 | | 2 | 1 |
| | 1766 | 301 | | 86 | 21 | | | |

Figure 9: The *standard query graphs*, along with the dynamic range (DR) of the counts of target graphs $G_t$ and the dynamic range of the canonical counts of neighborhoods $G_c$. The statistics are from the ENZYMES dataset.

| $G_q$ | DR $G_t$ | DR $G_c$ | $G_q$ | DR $G_t$ | DR $G_c$ | $G_q$ | DR $G_t$ | DR $G_c$ |
|---|---|---|---|---|---|---|---|---|
| | 3573 | 688 | | 1631 | 453 | | 6 | 5 |
| | 615 | 150 | | 804 | 318 | | 16 | 16 |
| | 5223 | 1035 | | 2071 | 732 | | 3 | 3 |
| | 1543 | 448 | | 526 | 330 | | 9 | 8 |
| | 4049 | 930 | | 41 | 17 | | | |
| | 1725 | 595 | | 928 | 296 | | | |

Figure 10: The *large query graphs*, and the dynamic range (DR) of counts of the whole target graphs $G_t$ and canonical neighborhoods $G_c$ in ENZYMES.

### D.3 HYPER-PARAMETER CONFIGURATIONS

**DeSCo configurations**. For DeSCo's canonical partition stage, we set $d = 4$ for all the tasks according to Theorem 1. For DeSCo's neighborhood counting stage, it contains two GNNs to encode target and query graphs into embedding vectors, and a regression model to predict canonical count based on the vectors. For the GNN encoders, we use the triangle-based message passing variant of GraphSAGE as shown in Table 8. The SHMP GNN has 8 layers with a feature size of 64. The canonical node of the neighborhood is marked with a special node type. The adjacent matrix $A$ is used to find the triangle and define the heterogeneous edge type with Equation 18.

$$E_{trianle} = \{(i,j)|(A \odot A^2)_{ij} > 0\} \tag{18}$$

For the Multilayer perceptron (MLP) of neighborhood counting, we use two fully-connected linear layers with 256 hidden feature size and LeakyReLu activation.

For the gossip correction stage, we use a two-layer GNN with 64 hidden feature size and a learnable gate as described in Equation 7. The learnable gate is a two-layer, 64-hidden-size MLP that takes the query embedding vector from the neighborhood counting stage and outputs the gate values for each GNN layer. The neighborhood counting prediction is expanded to 64 dimensions with a Linear layer and concatenated with the query embedding as the input for the two-layer GNN.

**Neural baseline configurations**. We follow the configurations of the official implementations of neural baselines and adapt them to our settings. They both contain two GNN encoders and a regression model like DeSCo's neighborhood counting model.

For LRP, we follow the official configurations for the ZINC dataset to use a deep LRP-7-1 graph embedding layer with 8 layers and hidden dimension 8. The regression model is the same as DeSCo.

For DIAMNet, we follow the official configurations for the MUTAG dataset to use GIN with feature size 128 as GNN encoders. The number of GNN layers is expanded from 3 to 5. The regression model is DIAMNet with 3 recurrent steps, 4 external memories, and 4 attention heads.

**Training details**. We use $C_c \leftarrow \log_2(C_c + 1)$ normalization for the ground truth canonical count $C_c$ to ease the high dynamic range problem. When evaluating the MSE of predictions, $\hat{C}_c \leftarrow 2^{\hat{C}_c} - 1$

is used to undo the normalization. We use the SmoothL1Loss with $\beta = 1.0$ from PyTorch Paszke et al. (2019) as the loss function to perform the regression task of neighborhood counting and gossip correction.

$$\mathcal{L}(\hat{C}_c, C_c) = \begin{cases} 0.5(\hat{C}_c - C_c)^2 & |\hat{C}_c - C_c| < 1 \\ |\hat{C}_c - C_c| - 0.5 & \text{otherwise} \end{cases} \tag{19}$$

We use the Adam optimizer for neighborhood counting and gossip correction and set the learning rate to 0.001. We align the computational resources when training different neural methods. DeSCo and DIAMNet have similar training efficiency, so DeSCo's neighborhood counting model and DI-AMNet are both trained for 300 epochs. After training the neighborhood counting model, DeSCo's gossip correction model is trained for 50 epochs with little resource consumption. In contrast, LRP is much slower. Even given twice training time, it can only be trained for 50 epochs.

**Approximate heuristic configurations**. For the MOTIVO baseline, we follow the official setting and use $10^7$ samples for each dataset. If the dataset has many graphs, the samples are evenly distributed on each target graph.

## E   RUNTIME COMPARISION

We use Intel Xeon Gold 6226R CPU with 2.90GHz frequency and NVIDIA GeForce RTX 3090 GPU for runtime tests.

**Method configurations**. For the exact method VF2 Cordella et al. (2004), we use the Python implementation from the graph processing framework Hagberg et al. (2008) and use Python's concurrent standard library to enable multiprocessing on four CPU cores. For the exact method IMSM Sun & Luo (2020), we use the official c++ implementation with four CPU cores. We use the IMSM-recommended method configurations: GQL He & Singh (2008) as the filtering method, RI Bonnici et al. (2013) as the ordering method, and LFTJ Bhattarai et al. (2019); Han et al. (2019) as the enumeration method. The failing set pruning optimization is also enabled. For the heuristic approximate method MOTIVO Bressan et al. (2019), we use the official c++ implementation with four CPU cores. For the neural method DeSCo, we use the Python implementation with one CPU core and one GPU core.

**Experiment setup**. All the methods are set to count the induced subgraphs in the ENZYMES dataset. Note that IMSM can only perform non-induced subgraph counting. So VF2, MOTIVO, and DeSCo are set to perform induced subgraph counting tasks, while IMSM performs non-induced tasks for runtime comparison. For query sizes no larger than five nodes, the *standard queries* from Section 5.1 are used. For query sizes larger than five, the same thirty queries of each size are selected for VF2 and DeSCo. We cannot assign specific queries for MOTIVO, so it is set to output the count of any thirty queries of each size.

**Runtime results**. Figure 11(a) shows the runtime of each method with four minutes' time-bound. The data loading and graph format conversion time is ignored for all methods. For the exact methods, VF2 and IMSM, the runtime grows exponentially because of the exponentially growing possible matches of each query size. For the approximate heuristic method MOTIVO, the exponential growth is mostly because of the build-up phase before sampling. While greatly reducing the sampling space, MOTIVO's build-up phase needs to color the nodes of the target graph and count colorful trees for each node, which has exponential time complexity with regard to the query size. For DeSCo, the queries and targets are independently embedded, and queries are relatively small compared with targets. Thus, DeSCo can easily scale for large queries. For example, scaling the query size from 3 to 13 only poses 57% additional overhead on the total runtime. We further extend the time budget for MOTIVO to 60 minutes and find that DeSCo achieves $15\times$, $53\times$, and $120\times$ speedup over MOTIVO for size 13 to 15 queries, respectively. As Figure 11(b) shows, currently DeSCo's triangle finding in *neighborhood counting* takes the majority of the runtime, which can be easily substituted with other efficient implementations, e.g., Donato et al. (2018), to further speed up DeSCo.

**Asymptotic complexity**. For the proposed DeSCo's three-step pipeline, assuming the average canonical neighborhood $G_n$ of the target graph $G_t$ has $V_n$ nodes and $E_n$ edges. The time complexity for canonical partition is the index-restricted breadth-first search starting from all the target

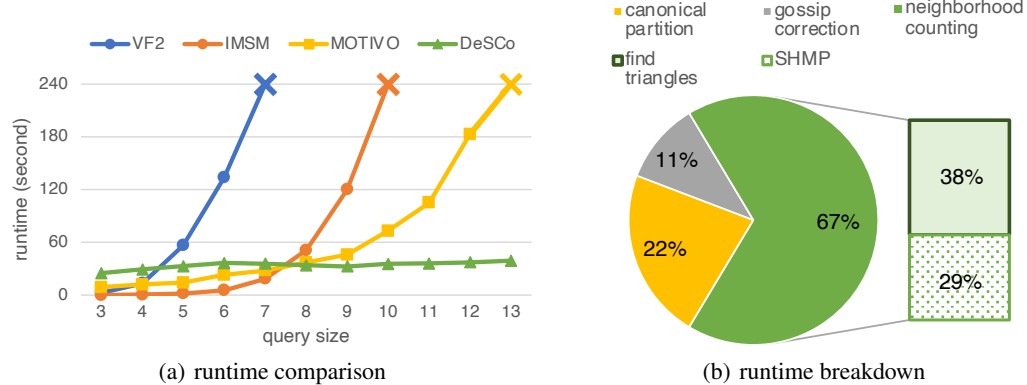

(a) runtime comparison  (b) runtime breakdown

Figure 11: The runtime comparison of different size queries and the runtime breakdown of DeSCo

vertices as shown in Appendix A.3, which is $O(V_t \times (\bar{V}_n + \bar{E}_n))$. The time complexity for neighborhood counting consists of triangle counting and heterogeneous message passing on $G_q$ and $G_t$. The complexity of triangle counting is $O(E^{3/2})$ on the target and query graph Itai & Rodeh (1977). The heterogeneous message passing has the complexity of regular GNNs Maron et al. (2019) on the $V_t$ neighborhoods and the queries, which is $O(V_t \times (\bar{V}_n + \bar{E}_n)) + O(V_q + E_q)$. For gossip correction, the time complexity also equals to a regular GNN, which is $O(E_t + V_t)$.

So the overall time complexity of DeSCo is $O(E_t^{3/2} + V_t \times (\bar{V}_n + \bar{E}_n)) + O(E_q^{3/2} + V_q)$. In real-world graphs, the common contraction of neighborhoods Weber (2019) makes $\bar{V}_n$ and $\bar{E}_n$ relatively small. So the major asymptotic complexity comes from the triangle counting on the target graph, which only has polynomial time complexity.

In contrast, for the heuristic approximate method MOTIVO, the build-up phase alone has time complexity $O(a^{V_q} \times E_t)$ for some $a > 0$. So it suffers from exponential runtime growth. For exact method VF2, the time complexity is $O(V^2)$ to $O(V! \times V)$ where $V = \max\{V_t, V_q\}$. In practice, we generally observe exponential runtime growth. Experiments of Figure 11 confirms the above analysis.

## F  COUNT DISTRIBUTION PREDICTION

To the best of our knowledge, DeSCo is the first approximate method that predicts the subgraph count distribution over the whole target graph. We use the canonical count of each node as the ground truth for the distribution prediction accuracy analysis. The canonical count represents the number of *pattern*s in each node's neighborhood while avoiding missing or double counting as discussed in Section 4.1. Following the setup in Section 5.1, we use all the size $3-5$ *standard query* graphs to test the distribution performance of DeSCo on different target graphs. The normalized MSE is the mean square error of the canonical count prediction of each (query, target graph node) pair divided by the variance of the (query, target graph node) pair's true canonical count. The MAE is the mean absolute error of the canonical count prediction of each (query, target graph node) pair.

| Dataset | MUTAG | | | COX2 | | | ENZYMES | | |
|---|---|---|---|---|---|---|---|---|---|
| Query-Size | 3 | 4 | 5 | 3 | 4 | 5 | 3 | 4 | 5 |
| Norm. MSE | 7.51E-2 | 2.36E-1 | 1.71E+0 | 4.94E-4 | 5.63E-4 | 1.44E-2 | 4.74E-5 | 5.69E-5 | 5.66E-4 |
| MAE | 2.97E-4 | 1.19E-3 | 1.90E-2 | 3.90E-4 | 5.19E-4 | 1.71E-2 | 7.60E-2 | 1.51E-1 | 3.15E-1 |

Table 8: DeSCo's count distribution prediction error under normalized MSE and MAE. Use the canonical count of each target graph node as the ground truth.

Experiments show DeSCo achieves a low $0.23$ normalized MSE for the count distribution prediction task. A visualization of DeSCo's distribution prediction on the CiteSeer dataset is also shown in

Figure 1. Note how DeSCo accurately predicts the distribution while providing meaningful insight on the graph.

# G    ADDITIONAL RESULTS ANALYSIS FOR LARGE QUERIES

To give an even more in-depth understanding of the performance for large queries, we additionally provide the results with more evaluation metrics.

## G.1    Q-ERROR ANALYSIS

**Definition**. Given the ground truth subgraph count $\mathcal{C}$ of query $G_q$ in target $G_t$, as well as the estimated count $\hat{\mathcal{C}}$. We use the definition of q-error from previous work Zhao et al. (2021).

$$e_q(G_q, G_t) = \max\left\{\frac{\mathcal{C}}{\hat{\mathcal{C}}}, \frac{\hat{\mathcal{C}}}{\mathcal{C}}\right\}, e_q \in [1, +\inf) \tag{20}$$

The q-error quantifies the factor that the estimation differs from the true count. The more it is close to 1, the better estimation. In Zhao et al. (2021), there is also an alternative form of q-error used in figures to show the systematic bias of predictions.

$$e_q(G_q, G_t) = \frac{\hat{\mathcal{C}}}{\mathcal{C}}, e_q \in (0, +\inf) \tag{21}$$

We follow the previous settings and use Equation 21 in our visualization.

**Experimental results**. We reassess the performance of DeSCo on large queries from Figure 5, and show the box plot in Figure 12. The data that $\mathcal{C} = 0$ is ignored for mathematic correctness. The box of MOTIVO on MUTAG is too close to zero to be shown in the figure. DeSCo's q-error is the closest to 1 with minimal spread. It shows how DeSCo excels in systematic error and consistency compared with the baselines.

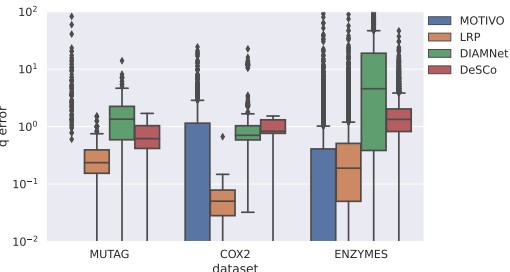

Figure 12: The q-error box plot of large query-target pairs. The q-error (y-axis) is clipped at $10^{-2}$ and $10^2$. For q-error, the closer to 1, the better.

**Limitations of q-error**. Despite the advantage of demonstrating relative error, the q-error metric also has obvious limitations, thus not being chosen as the major evaluation metric. In Zhao et al. (2021), the authors assume $\mathcal{C} \geq 1$ and $\hat{\mathcal{C}} \geq 1$. However, this assumption may not hold, given that the query graph may not (or is predicted to) exist in the target graph, especially for larger queries. The zero or near-zero denominators greatly influence the average q-error. It causes the overestimation of the subgraph existence problem instead of the subgraph counting problem.

## G.2    MSE ANALYSIS

**Definition**. We follow the same setting in Figure 5 and show the normalized MSE for predicting the subgraph count of large queries. Note that in a few cases, the tested large queries of a certain size may not exist in the target graph. For example, the two size-thirteen queries in Figure 10 do not

exist in the CiteSeer dataset. To prevent divide-by-zero in normalization, the MSE is normalized by the variance of ground truth counts of all large queries, instead of being normalized for each query size.

**Experimental results**. The experimental results are shown in Table 9. DeSCo demonstrates the lowest MSE on all tested target graphs.

| Dataset | MUTAG | COX2 | ENZYMES |
|---------|-------|------|---------|
| MOTIVO | 1.2E+01 | 3.2E+00 | 1.4E+00 |
| LRP | 7.6E-01 | 1.1E+00 | 6.7E-01 |
| DIAMNet | 2.3E+00 | 2.4E-01 | 8.9E+01 |
| DeSCo | **1.5E-01** | **1.2E-01** | **4.0E-01** |

Table 9: Normalized MSE of approximate heuristic and neural methods on subgraph counting of sixteen large queries.

