# OpenReview forum: "DeSCo: Towards Scalable Deep Subgraph Counting"
_ICLR.cc/2023/Conference — Submitted to ICLR 2023_

### Official Review · Reviewer_Gava · 2022-10-22

**Confidence:** 3
**Correctness:** 3
**Technical Novelty And Significance:** 2
**Empirical Novelty And Significance:** 2
**Recommendation:** 3

**Clarity, Quality, Novelty And Reproducibility:**

The clarity is good, but the quality and novelty are less satisfactory.

The reproducibility could be better supported by showing the robustness of performance.


**Strength And Weaknesses:**

Strength:

The proposed ideas are reasonable. The identified example for showing the expressiveness of SHMP is particularly interesting. Most of the sections are well-written and easy to follow. Source codes are provided through external links.


Weaknesses:

- W1: The proposed method is reasonable but does not address the fundamental challenge in subgraph counting: the problem is still computationally hard after fixing a pair of nodes or a certain order of the nodes. That is, the acquired subproblems are in general as hard as the original problem. In this regard, the methods of canonical partition and gossip correction seem simplistic.

- W2: The theoretical contribution is not strong enough, as there is no formal analysis other than one example. It would be better to justify the model expressiveness in a formal way. In addition, the results in Sec 4.2 are pretty straightforward.

- W3: The experiments could be improved in the following aspects:
  - Given the nature of the proposed method, the performance seems to depend on the node indexing schemes. Such an issue should be investigated at least in experiments.
  - To justify the statistical significance of the learning performance, one should provide the robustness over the testing set (e.g., std) as well as the robustness over multiple individual trainings (e.g., cross-validation), which are required to rule to out the cases where a) the average error is low but the standard deviation is overly high and b) the trained model is accidentally good.
  - To make it possible to have an in-depth understanding of the results, it is better to show the exact counting results rather than MSE.
  - The hyperparameter P seems important, and therefore, it would be better to carefully study its impacts in experiments.
  - One of the common practices is to select fixed query graphs, and then examine the performance; this paper lists a collection of query graphs, but it is not completely clear to me how they form the training-testing sets. Since this paper involves existing methods in experiments, it could be better to follow the exact settings in those papers, which will make the comparison clearer.
  -The paper mentions pretraining but not training, which is a little bit confusing.

- W4: Some minor comments:
  - What does it mean by “combinatorially long runtime”?
  - The paper claims that their approach can easily support graphs with node features, which, however, seems non-trivial. A small experiment can be included to support such a claim.
  - Some notations in Sec 4.1 are not well-defined.


**Summary Of The Paper:**

This paper studies the problem of solving the subgraph counting problem using approaches based on neural networks. The proposed method leverages canonical partition to decompose the considered problem into subproblems, and the entire counting is further calibrated by gossip correction balancing homophily and antisymmetry. Experimental studies are provided to support the proposed designs.


**Summary Of The Review:**

This paper has some interesting ideas, but the overall contribution is not significant. The experiments need to be more convincing.

---

> ### Author Response · Authors · 2022-11-17
> **Author Response (Minor Comments)**
>
> **W4: Minor Comments**
>
> **W4.1: Meaning of "combinatorially long runtime"?**
>
> > What does it mean by "combinatorially long runtime"?
>
> It means the asymptotic runtime is $O(C(V_t,V_q))$, where C is the combination formula $C(n,k)=\frac{n!}{k!(n-k)!}$. The runtime of exact subgraph counting can be combinatorial because it can be done by iteratively selecting the same rows $\{i_1,\dots,i_{V_q}\}$ and columns $\{i_1,\dots,i_{V_q}\}$ from the $V_t \times V_t$ target adjacency matrix to form the $V_q \times V_q$ query adjacency matrix. We thank the reviewer for the question and will further clarify it in the revision.
>
> **W4.2: Support Node Features**
>
> > The paper claims that their approach can easily support graphs with node features, which, however, seems non-trivial. A small experiment can be included to support such a claim.
>
> Our embedding method is GNN-based, thus can naturally support input node features. One can simply do so by replacing the trivial zero vector  $x_i^{(0)}$ to the node feature vector in Equation 6 and Equation 7.
>
> **W4.3: Notations in Sec 4.1**
>
> > Some notations in Sec 4.1 are not well-defined.
>
> We thank the reviewer and will clarify the notations in the revision.

---

> ### Author Response · Authors · 2022-11-17
> **Author Response (Experiments)**
>
> **W3: Experiments**
>
> We note that most suggestions are factors already considered/mentioned in the original paper. We did add minor additional information for some of the concerns.
>
> **W3.1: Influence of the Node Indexing Schemes**
>
> > Given the nature of the proposed method, the performance seems to depend on the node indexing schemes. Such an issue should be investigated at least in experiments.
>
> We emphasize that the influence of the node indexing scheme is limited by nature. As shown in Definition 4.1, the canonical count definition only reallocates the count inside a small range, the pattern $G_p$, whose size is small (equals to the query graph). Figure 8 also shows that the correlation $r$ between the node index and the canonical count is below 0.45. Given the limited influence, we use the random node indexing scheme to achieve average performance. Yet we agree that the node indexing scheme may slightly influence the performance. We will add the experiments of different schemes, e.g., random and order-by-degree, in our final revision.
>
> **W3.2: The Robustness Over the Testing Set**
>
> > To justify the statistical significance of the learning performance, one should provide the robustness over the testing set (e.g., std) as well as the robustness over multiple individual trainings (e.g., cross-validation), which are required to rule to out the cases where a) the average error is low but the standard deviation is overly high and b) the trained model is accidentally good.
>
> We thank the reviewer for the suggestion and will add the statistics shortly.
>
> **W3.3: Use Exact Counting Results as the Evaluation Metric**
>
> > To make it possible to have an in-depth understanding of the results, it is better to show the exact counting results rather than MSE.
>
> The above statement is, in fact, impractical. We cannot show the exact counting numbers since there are too many of them: 1255 target graphs multiply by 29 standard queries for just Table 3. However, as pointed out by reviewer Eqk2, we have shown the results in various metrics and detailed distributions to give the readers a more in-depth understanding. For example, the accumulative error distribution in Figure 5, and the q-error distribution in Figure 12.
>
> **W3.4: Study Hyperparameter P**
>
> > The hyperparameter P seems important, and therefore, it would be better to carefully study its impacts in experiments.
>
> It is a misunderstanding: the learnable gate P is not a hyperparameter but a learned variant for different queries. We clearly state that "P is learned from the query embedding" in Section 4.4. We also state the details that "the learnable gate is a two-layer, 64-hidden-size MLP that takes the query embedding vector from the neighborhood counting stage and outputs the gate values for each GNN layer" in Appendix D.3.
>
> **W3.5: Explain How to Choose Queries**
>
> > One of the common practices is to select fixed query graphs, and then examine the performance; this paper lists a collection of query graphs, but it is not completely clear to me how they form the training-testing sets. Since this paper involves existing methods in experiments, it could be better to follow the exact settings in those papers, which will make the comparison clearer.
>
> We have detailed the choices of queries and the training method in the original submission. The set of "standard query graphs" used in most experiments "include all non-isomorphic, connected, undirected, graphs with node size 3 − 5" (Section 5.1). The list of all the "standard query graphs" are shown in Figure 8. For training, we train "all the neural methods with the target-query pairs from the synthetic dataset and the standard queries of size 3 − 5" (Section 5.1). For testing, "unless specified, the trained model is directly tested with standard queries" and "the targets from unseen, real-world datasets" (Section 5.1). The statistics of the datasets are shown in Table 1. "All the neural methods" (Section 5.1) are trained and tested using the same query set, the same target set, and the same training scheme. We believe it provides a clear comparison between baselines.
>
> **W3.6: Explain the Term 'pre-training'**
>
> > The paper mentions pre-training but not training, which is a little bit confusing.
>
> The term 'pre-training' means the neural subgraph counting model is trained once with a large and comprehensive dataset, then directly used for inference on real-world datasets from various domains without further training. It poses higher requirements on the generalization and expressive power of the model, since it requires adaptivity to all kinds of data distributions. Under such an objective, we make the neural method a practical substitute for existing heuristic counting methods out-of-the-box, only faster, more accurate, and more scalable.

---

> ### Author Response · Authors · 2022-11-17
> **Author Response (W2)**
>
> **W2: Theoretical Contribution Not Strong Enough**
>
> > The theoretical contribution is not strong enough, as there is no formal analysis other than one example. It would be better to justify the model expressiveness in a formal way.
>
> The theoretical contribution is twofold. First, we prove the stronger expressive power of SHMP. Note that constructing a counter-example is only part of the proof, and is commonly used to prove the expressive power in previous work, e.g., [2]. The complete proof works in three stages. Firstly, we construct an SHMP-GIN that is at least as powerful as the 1-WL test. Then we construct a d-regular graph counter-example to show that this SHMP-GIN can distinguish certain graphs that the 1-WL test cannot, proving strictly stronger expressive power. Finally, since previous work [4] has shown that the expressive power of existing MPGNNs is upper-bounded by the 1-WL test, we can assert that SHMPGNN's "expressive power can exceed the upper bound of that of MPGNNs."
>
> Second, for the canonical partition, we theoretically prove the equivalence between the sum of the canonical count and the subgraph count. The canonical count allows the use of canonical partition to decompose the \#P hard counting problem into multiple simple subproblems. We additionally prove that such decomposition prevents missing or double counting. Canonical partition helps to fundamentally address the subgraph counting problem as discussed in W1.
>
> We would like to emphasize that this is a practical and novel approach to subgraph counting, but not a theoretical paper on counting. Compared to previous works on neural approaches, we have already provided abundant theoretical justification and analysis. We would like to point out that our focus is instead on the architecture design and the gossip counting, which conforms to the inductive biases of practical subgraph counting problems.
>
> [4] Keyulu Xu, Weihua Hu, Jure Leskovec, and Stefanie Jegelka. How powerful are graph neural networks? *arXiv preprint arXiv:1810.00826*, 2018.

---

> ### Author Response · Authors · 2022-11-17
> **Author Response (W1)**
>
> **W1.1:  Subproblems After Canonical Partition are Still As Hard As the Original Problem**
>
> > The proposed method is reasonable but does not address the fundamental challenge in subgraph counting: the problem is still computationally hard after fixing a pair of nodes or a certain order of the nodes. That is, the acquired subproblems are in general as hard as the original problem. In this regard, the methods of canonical partition and gossip correction seem simplistic.
>
> We would like to disagree with the comment politely. The canonical partition fundamentally addresses the problem in three aspects:
>
> 1. Search space reduction.
>
>    We denote the complexity $\text{S}$ of counting on graph $G_t$ as $\text{S}(V_t)$. Canonical partition divides the $\text{S}(V_t)$ complex original problem into subproblems with the total complexity of $\sum_{i \in V_t} \text{S}(V_{n_i})$. Thanks to the sparse nature of real-world graphs, $V_{n_i}$s are generally small, even for huge targets. We use different $S$ ( $\text{S(V)}=O(V!\times V)$ for VF2's worst case; $\text{S}(V)=2^V$ for empirical observation) to estimate the worst and average search space on real-world graphs. Statistics show that the search space is drastically reduced by a factor of $1/10^{70}$ and $1/10^{11}$, respectively.
>
> 2. Redundant match elimination
>
>    Canonical partition, along with the canonical count definition, eliminates the redundant automorphic match of the query graph. It transforms the $\mathbb{R}^3 \mapsto \mathbb{R}^3$ bijection $f$, the query-target mapping, to the $\mathbb{R}^3 \mapsto \mathbb{R}$ bijection $f_c$. Up to $V_q!-1$ redundant matches can be eliminated.
>
> 3. Reduction in the dynamic range of labels
>
>     Canonical partition also reduces the subgraph count's dynamic range, making the regression task easier for the neural network.
>
> Compared to previous literature [1~3] on neural subgraph counting, our approach is the first step to substantially reduce the problem complexity in the neural framework. The discussion has been detailed in Appendix A.4 in the revision.
>
> Additionally, the experimental results in Table 4 also show drastic MSE reductions with the help of canonical partition, further confirming the method's effectiveness. We thank the reviewer for raising the question. The paper is more convincing with the added analysis and discussions.
>
> [1] Xin Liu, Haojie Pan, Mutian He, Yangqiu Song, and Xin Jiang. Neural subgraph isomorphism counting. Proceedings of the 26th ACM SIGKDD International Conference on Knowledge Dis- covery & Data Mining, 2020.
>
> [2] Zhengdao Chen, Lei Chen, Soledad Villar, and Joan Bruna. Can graph neural networks count substructures? ArXiv, abs/2002.04025, 2020.
>
> [3] Kangfei Zhao, Jeffrey Xu Yu, Hao Zhang, Qiyan Li, and Yu Rong. A learned sketch for subgraph counting. Proceedings of the 2021 International Conference on Management of Data, 2021.
>
> **W1.2: Subproblems After Gossip Correction are Still Hard**
>
> We clarify that the gossip correction does not relate to the complexity of the subproblems. Instead, it is used to "improve the prediction quality" by "modeling the error of neighborhood count." It "enforces different homophily and antisymmetry inductive biases for different queries," which is not simplistic and has never been seen in previous works to the best of our knowledge.

---

### Official Review · Reviewer_eb8f · 2022-10-24

**Confidence:** 4
**Correctness:** 3
**Technical Novelty And Significance:** 3
**Empirical Novelty And Significance:** 3
**Recommendation:** 6

**Clarity, Quality, Novelty And Reproducibility:**

Clarity: The paper is clear overall. The problem, competing methods, proposed approach, and experiments are clear to me.
Quality: The manuscript would benefit from proof-reading to fix the numerous typos, which start at the first sentence in the abstract and end at the next-to-last sentence in the conclusions. The technical quality is below average.
Originality: In general terms, the idea of breaking down the subgraph counting problem into subproblems has been explored before (see, for instance, all the work in color-coding algorithms). However, I believe that there is novelty in the particular way in which the authors partition the graph.


**Strength And Weaknesses:**

Strengths:
- Improved performance for a well-studied, relevant problem in the graph theory / network analysis literature
- There is some novelty on the proposed SHMP module
- Paper is well-organized and the visuals are very useful for understanding the proposed method

Weaknesses:
- It is not clear how the improved performance in MSE translates to better results in practice. Why should a practitioner care that the proposed method is more accurate? Is there, for instance, a case study where the proposed approach leads to new biological insights?
- It is not clear why the authors decided to present squared error as a distribution in Figure 5. I think it would be more informative to see raw squared error numbers, as they present in the rest of the paper.
- No comparison in terms of runtime to DIAMNet
- Many typos across the paper

**Summary Of The Paper:**

The authors propose a deep learning model for the task of subgraph counting. This is a challenging algorithmic problem and a well-known #P problem with applications in biology and social science. Deep learning approaches for subgraph counting have been proposed before, but the model presented by the authors improves both in the accuracy and, most notably, in scalability over existing approaches.

**Summary Of The Review:**

See my comments above. Overall, I believe this paper would make a fine addition to the conference and to the subgraph counting literature.

---

> ### Author Response · Authors · 2022-11-17
> **Author Response 2**
>
> **W3: Runtime Comparison to DIAMNet**
>
> > No comparison in terms of runtime to DIAMNet
>
> We point out that neural methods DIAMNet and DeSCo "have similar training efficiency" (Section D.3) in terms of runtime. Unlike exact and heuristic approximate methods, the biggest challenge for neural methods is accuracy rather than runtime efficiency. We emphasize that our method achieves accuracy improvement by two orders of magnitude in terms of MSE, while preserving the runtime benefit of neural baselines. We also value the reviewer's advice and will add the runtime data of neural methods in the revision.
>
> **W4: Typos**
>
> > Many typos across the paper
> > The manuscript would benefit from proof-reading to fix the numerous typos
>
> We genuinely thank the reviewer for pointing out the typo problem. We carefully examined the paper. After fixing the typos, we think this will no longer be a concern.
>
> **Discussion: Originality**
>
> > Originality: In general terms, the idea of breaking down the subgraph counting problem into subproblems has been explored before (see, for instance, all the work in color-coding algorithms). However, I believe that there is novelty in the particular way in which the authors partition the graph.
>
> We are glad that the reviewer appreciates the novelty of the canonical partition. We would also like to emphasize the originality of the subgraph-based heterogeneous message passing (SHMP) and the gossip correction model. SHMP boosts the expressive power of GNNs with higher efficiency than previous works. The gossip correction model uses the learnable gate to balance the mutually exclusive homophily and antisymmetry properties of the counting problem. The methods provide insight that has never been seen before in previous papers.

---

> ### Author Response · Authors · 2022-11-17
> **Author Response 1**
>
> **W1: Influence of Accuracy Improvement on Downstream Tasks**
>
> > It is not clear how the improved performance in MSE translates to better results in practice. Why should a practitioner care that the proposed method is more accurate? Is there, for instance, a case study where the proposed approach leads to new biological insights?
>
> We wish to clarify a potential misunderstanding: this work aims to improve the accuracy of subgraph counting itself, not the downstream tasks. Accurate and faster subgraph counting is the pursuit of the long line of works in the domain. The survey [2] discussed 16 approximate counting algorithms from the year 2004 to 2018, all of which are evaluated by count estimation accuracy and runtime efficiency, not the improvement on downstream tasks. Hence we are not creating an artificial task, but are instead using neural approaches to an already established problem setting.
>
> Additionally, we would like to refer to the previous work DIAMNet [1] to show the significance of our accuracy improvement. We use the MSE metric for both our work and DIAMNet. DIAMNet's experiments show a maximum $5.1\times$ improvement over the worst CNN baseline, and $11.4\times$ improvement over the "always-zero" trivial prediction. In comparison, our work demonstrates an additional $437\times$ improvement over DIAMNet, which is much more significant even with a stronger baseline.
>
> [1] Xin Liu, Haojie Pan, Mutian He, Yangqiu Song, and Xin Jiang. Neural subgraph isomorphism counting. *Proceedings of the 26th ACM SIGKDD International Conference on Knowledge Discovery & Data Mining*, 2020.
>
> [2] Pedro Ribeiro, Pedro Paredes, Miguel EP Silva, David Aparicio, and Fernando Silva. A survey on subgraph counting: concepts, algorithms, and applications to network motifs and graphlets. ACM Computing Surveys (CSUR), 54(2):1–36, 2021.
>
> **W2: Why use Error Distribution Instead of MSE for Large Queries**
>
> > It is not clear why the authors decided to present squared error as a distribution in Figure 5. I think it would be more informative to see raw squared error numbers, as they present in the rest of the paper.
>
> We would like to point out a clear justification for using the metric: it is more comprehensive, and more illustrative for in-depth analysis, compared to MSE. The error distribution can better demonstrate the performance for small and very large ground truth counts, which are both crucial for large queries. The count distribution of large queries is sparse and screwy: most large queries have zero or small counts, while few large queries have very large counts. The MSE can emphasize too much on the errors of the large outlier counts, while the count distribution can show both. For example, Figure 5(b) shows that the LRP method has more valid predictions than DIAMNet for small error bounds, which means most of LRP's predictions are accurate. However, LRP's validity does not improve much given the larger error bound, meaning that few of LRP's predictions have very large errors. This confirms the q-error analysis in Figure 11, where LRP often makes zero or smaller count predictions to fit most queries, while failing to predict the large count of few queries frequently seen in the target graph.
>
> It is clear that distribution provides more information than the mean error. Yet we thank the reviewer for the question and will add the MSE numbers in the appendix for completeness.

---

### Official Review · Reviewer_Eqk2 · 2022-10-27

**Confidence:** 4
**Clarity, Quality, Novelty And Reproducibility:** The paper is well written. There is s…
**Correctness:** 3
**Technical Novelty And Significance:** 3
**Empirical Novelty And Significance:** 2
**Recommendation:** 6

**Strength And Weaknesses:**

Strength:

1. Experimental analysis (including those in appendix) is extensive. Additional scalability study / q-error analysis present new insights.

2. The idea of canonical partitioning is novel to me.

Weakness:

1. Apart from exact methods, there are also many recent sampling based estimation methods. Sampling-based methods are not exact, but are significantly faster and often have theoretical guarantees. Some discussion or even comparison to this line of work is needed.

[1] Bressan, Marco, Stefano Leucci, and Alessandro Panconesi. "Faster motif counting via succinct color codingand adaptive sampling." ACM Transactions on Knowledge Discovery from Data (TKDD) 15.6 (2021): 1-27.
[2] Wang, Pinghui, et al. "Efficiently estimating motif statistics of large networks." ACM Transactions onKnowledge Discovery from Data (TKDD) 9.2 (2014): 1-27.

2. Traditional methods like VF2 runs on CPU, which is not exactly a fair comparison to GNN methods which runs on GPUs. There are also some GPU-based exact methods [3]

[3] Lin, Wenqing, et al. "Network motif discovery: A GPU approach." IEEE transactions on knowledge and dataengineering 29.3 (2016): 513-528.

3. There are also some newer GNN-based baseline [4].

[4] Liu, Xin, and Yangqiu Song. "Graph convolutional networks with dualmessage passing for subgraph isomorphism counting and matching." Proceedings of the AAAI Conference onArtificial Intelligence. 2022.

4. It would be more beneficial if the authors can demonstrate some case work--- how reduced error translate to accuracy in downstream tasks that makes use of such counts? While it is certainly good that MSE is reduced, e.g. from 1 to 0.6, it is not intuitive how significant is such reduction.

**Summary Of The Paper:**

In this work, the authors designed a GNN-based framework for subgraph counting. Previous exact/heuristic methods are often slow, given  an NP hard problem. There are several key ideas: 1. canonical partition, 2. heterogeneous message passing, and 3. gossip correction. Extensive experiments are conducted to evaluate the effectiveness of the proposed approach.

**Summary Of The Review:**

In short, i think the strengths slightly outweigh the weaknesses.

---

> ### Author Response · Authors · 2022-11-17
> **Author Response 2**
>
> **W3: Additional GNN Baseline**
>
> > There are also some newer GNN-based baseline [4].
>
> We thank the reviewer for providing the additional GNN baseline. We are working on the comparison with it in the final revision. For now, we suspect that [6] may not scale efficiently to large queries. It requires the analysis of the corresponding line graph of the query, which has maximally $n(n-1)/2$ nodes (see Figure 3 in [6]). For example, 13-clique counting would require the analysis on a 78-node line graph, which seems less efficient.
>
> [6] Liu, Xin, and Yangqiu Song. "Graph convolutional networks with dualmessage passing for subgraph isomorphism counting and matching." Proceedings of the AAAI Conference on Artificial Intelligence. 2022.
>
> **W4: Influence of Accuracy Improvement on Downstream Tasks**
>
> > It would be more beneficial if the authors can demonstrate some case work--- how reduced error translate to accuracy in downstream tasks that makes use of such counts? While it is certainly good that MSE is reduced, e.g. from 1 to 0.6, it is not intuitive how significant is such reduction.
>
> We wish to clarify a potential misunderstanding: this work aims to improve the accuracy of subgraph counting itself, not the downstream tasks. Accurate and faster subgraph counting is the pursuit of the long line of works in the domain. The survey [4] discussed 16 approximate counting algorithms from the year 2004 to 2018, all of which are evaluated by count estimation accuracy and runtime efficiency, not the improvement on downstream tasks. Hence we are not creating an artificial task, but are instead using neural approaches to an already established problem setting.
>
> Additionally, we would like to refer to the previous work DIAMNet[7] to show the significance of our accuracy improvement. We use the MSE metric for both our work and DIAMNet. DIAMNet's experiments show a maximum $5.1\times$ improvement over the worst CNN baseline, and $11.4\times$ improvement over the "always-zero" trivial prediction. In comparison, our work demonstrates an additional $437\times$ improvement over DIAMNet, which is much more significant even with a stronger baseline.
>
> [7] Xin Liu, Haojie Pan, Mutian He, Yangqiu Song, and Xin Jiang. Neural subgraph isomorphism counting. *Proceedings of the 26th ACM SIGKDD International Conference on Knowledge Discovery & Data Mining*, 2020.

---

> ### Author Response · Authors · 2022-11-17
> **Author Response 1**
>
> **W1: Additional Approximate Baselines**
>
> > Apart from exact methods, there are also many recent sampling-based estimation methods. Sampling-based methods are not exact, but are significantly faster and often have theoretical guarantees. Some discussion or even comparison to this line of work is needed.
>
> We already compared with the state-of-the-art method [1] under the name "MOTIVO [2]" in our paper. We demonstrated excellent polynomial scalability, as discussed in Appendix E. Though early sampling-based methods like [3] can scale to large targets, none of them, to the best of our knowledge, efficiently scale to large queries (e.g., up to 13 nodes) as ours. As discussed in Table 3. of the survey [4], even state-of-the-art sampling-based methods can only support queries with no more than seven nodes. Our method's scalability is unprecedented from that perspective.
>
> We wish to further clarify the citation of MOTIVO. Reference [1] is an extended version of [2] that presents "two motif counting algorithms": "Motivo, which is designed to count motifs on $k \leq 16$ nodes," and "L8Motif, which is optimized for motifs on $k \leq 8$ nodes". The context of this paper is scalable subgraph counting, and we conduct runtime experiments on queries with up to 13 nodes; thus, L8Motif is not applicable, given its query size limit of 8. Both [1] and [2] refer to the same source code [repository](https://gitlab.com/steven3k/motivo/), so we follow the instructions in the repository to cite MOTIVO as [2] and add [1] to the reference.
>
> [1] Marco Bressan, Stefano Leucci, and Alessandro Panconesi. Faster motif counting via succinct color codingand adaptive sampling. ACM Transactions on Knowledge Discovery from Data (TKDD) 15.6 (2021): 1-27.
>
> [2] Marco Bressan, Stefano Leucci, and Alessandro Panconesi. Motivo: fast motif counting via succinct color coding and adaptive sampling. Proceedings of the VLDB Endowment, 12(11):1651–1663, 2019.
>
> [3] Wang, Pinghui, et al. "Efficiently estimating motif statistics of large networks." ACM Transactions on Knowledge Discovery from Data (TKDD) 9.2 (2014): 1-27.
>
> [4] Pedro Ribeiro, Pedro Paredes, Miguel EP Silva, David Aparicio, and Fernando Silva. A survey on subgraph counting: concepts, algorithms, and applications to network motifs and graphlets. ACM Computing Surveys (CSUR), 54(2):1–36, 2021.
>
> **W2: Additional Exact GPU Baseline**
>
> > Traditional methods like VF2 runs on CPU, which is not exactly a fair comparison to GNN methods which run on GPUs. There are also some GPU-based exact methods [5]
>
> We compare with the GPU-based exact methods [5] and find that our method can easily beat it. For example, when both are set to find size-4 to size-8 queries on the $10^4$-scale target graphs (CE and ENZYMES), [5] takes $1\times10^4$ seconds (Figure 4(a) in [5]), while our method only takes $1.7\times10^2$ seconds. Though the query graphs are not exactly the same, ours is two orders of magnitude faster in general. We wish to provide a more detailed and precise comparison; unfortunately, [5] does not open-source their code.
>
> We wish to further emphasize that the runtime comparison with CPU baselines is not unfair. As discussed in Appendix E of the original submission, they use "four CPU cores," while our method uses "one CPU core" and "one GPU." Additionally, the IMSM and MOTIVO baselines are implemented with "efficient c++" code and dedicated optimization on speed, while ours "uses the Python implementation."
>
> [5] Lin, Wenqing, et al. "Network motif discovery: A GPU approach." IEEE transactions on knowledge and dataengineering 29.3 (2016): 513-528.

---

### Author Response · Authors · 2022-11-18
**General Response**

We genuinely thank the reviewers for their valuable feedback. We are glad that they found our neural approach to subgraph counting "novel" and "notably improves scalability" with "well-written" content, "useful visuals," and "extensive experiments."
We appreciate that the reviewer thinks our method "would make a fine addition to the conference and to the subgraph counting literature." During the rebuttal, we aim to address the reviewers' comments while clarifying some misunderstandings. We also wish to solve the major concerns, e.g., how our method fundamentally reduces the counting complexity on real-world graphs. Finally, we hope the reviewer could raise the score if satisfied with our responses.

P.S. The major revision of the paper is marked with blue color for clarity.

---

### Decision · Program_Chairs · 2023-01-20

**Decision:**

Reject

**Justification For Why Not Higher Score:**

The paper shows improvements for a very narrow task and do not consider the most advanced techniques.

**Justification For Why Not Lower Score:**

N/A

**Metareview: Summary, Strengths And Weaknesses:**

The paper presents a new algorithm, DeSCo, for subgraph counting in large graphs. The new proposed algorithm works in three stages. It first builds a canonical partition to reduce the problem from the entire graph to small subgraphs. Then it uses a novel subgraph-based GNN approach to improve expressiveness and finally a gossip correction phase is used to improve the accuracy of the model.

Overall, the paper introduces some nice new ideas and it presents some interesting results but the paper still has some drawbacks that should be addressed before publication.

First, the experiments are not fully convincing and they are run only on a small dataset. As a comparison, the previous paper in the topic analyze graphs larger by 3 orders of magnitude.

Second, there have been important advances in the sampling based estimation methods(for example the one presented in Bressan "Efficient and near-optimal algorithms for sampling connected subgraphs". ACM STOC 2021) that are not discussed in the paper but that theoretically should lead to better accuracy.

Overall, the paper is interesting and contains some nice ideas but it is not ready for publication in the current state.